# Average annual costs of Rheumatoid Arthritis estimated by inverse probability weighting and their influence factors: A cross-sectional study based on Chinese Registry of Rheumatoid arthritis (CREDIT) Cohort

Bing Yu[1]⊗, Lin Qiao[2]⊗, Lu Li[1], Keying Zuo[1], Liangming Li[1], Li Wang[1], Nan Jiang[2], Qian Wang[2], Mengtao Li[2], Yanhong Wang[1]*, Xinping Tian[2]*

1 Department of Epidemiology and Biostatistics, Institute of Basic Medical Sciences Chinese Academy of Medical Sciences & School of Basic Medicine Peking Union Medical College, Beijing, China,
2 Department of Rheumatology and Clinical Immunology, Chinese Academy of Medical Sciences & Peking Union Medical College, National Clinical Research Center for Dermatologic and Immunologic Diseases (NCRC-DID), Ministry of Science & Technology, Key Laboratory of Rheumatology and Clinical Immunology, Ministry of Education, State Key Laboratory of Complex Severe and Rare Diseases, Peking Union Medical College Hospital (PUMCH), Beijing, China

⊗ These authors contributed equally to this work.
* wyhong826@pumc.edu.cn (YW); tianxp6@126.com (XT)

## Abstract

### Objective

To date, the evidences of economic burden for the RA individual in the real-world clinical practice were still limited in China. This study aimed to estimate average annual costs of rheumatoid arthritis (RA) patients using inverse probability weighting (IPW) and their influence factors.

### Methods

A multicenter, cross-sectional study was conducted and the RA patients who met inclusion criteria on CREDIT cohort were invited to participate the survey. After they signed the informed confirm form, the information of outpatient and inpatient expenditures in the past year were collected through online questionnaires. Medical records were retrieved from the Chinese Rheumatology Information System (CRIS). Propensity scores for sample using the Generalized Boosted Model (GBM) method were used to calculate IPW, so that we produced the weighted population similar to the target of RA patients on CREDIT. Bootstrap methods were used to estimate average costs and 95% confidence intervals with 1,000 samples. Indirect costs were estimated using the human capital approach. Weighted multivariate regression identified factors influencing average annual costs.

**Data availability statement:** The data that support the findings of this study are not publicly available due to ethical restrictions as participants did not consent to sharing their data publicly. The dataset is secured at National Clinical Research Center for Dermatologic and Immunologic Diseases (NCRC-DID) (https://www.ncrcdid.org.cn) at Peking Union Medical College Hospital (PUMCH), Beijing, China. Access to the data request can be applied for by shared mailbox: NCRC-DID@163.com or by emailing the coordinator via email (Luyu_doctor@163.com).

**Funding:** This study was supported by the Chinese National Key Technology R&D Program, Ministry of Science and Technology (2022YFC2504600, 2022YFC3601800), CAMS Innovation Fund for Medical Sciences (CIFMS) (2021-I2M-1-005, 2022-I2M-1-004, 2023-I2M-2-005), The Non-profit Central Research Institute Fund of Chinese Academy of Medical Sciences (2021-PT320-002, 2019-PT330-004), National High Level Hospital Clinical Research Funding (2022-PUMCH-B-013), The Special Science Research for Health Development in Capital (No.2024-1G-2082).

**Competing interests:** The authors have declared that no competing interests exist.

## Results

In this study, a total of 18,507 patients from the CREDIT database met the recruitment criteria. Among them, 1,293 patients from 152 hospitals across 29 provinces in China completed the questionnaire and were included in our analysis. The average annual total costs per patient by the Bootstrap method on the weighted population was about 41,971 CNY (Bootstrap 95%CI: 37,107–47,046 CNY), in which more than 75% were the direct costs. Moreover, in the direct costs, the medical costs accounted for nearly 89% and even half of them (approximately 59.7%) was medication expense. The moderate or high disease activity status, hospitalization in last year, history of comorbidity, the treatment of biologics or glucocorticoids were also found to substantially increase average annual costs of RA in our study.

## Conclusions

This study provided reliable insight into evaluating the economic burden of RA for the individuals and their families in the real-world clinical practice of China.

## Introduction

Rheumatoid arthritis (RA), a progressive inflammatory disease, often leads to joint damage and functional impairment associated with long-term pain and significant disabilities [1]. At global, it affected approximately 0.2–1.0% of the population, with notable regional variations [2]. In China, the prevalence of rheumatoid arthritis aligns closely with global estimates, whereas its incidence appears modestly higher. [3–4]. The most recent large-scale epidemiological study reported an overall prevalence of 368.11 per 100,000 people and an incidence rate of 140.64 per 100,000 person-years in 2017 of China, with a female-to-male ratio of 1.58:1 [4]. During the past decades, the burden of RA has been on a significant upward trend around the world with 7.4% increase in the global age-standardized prevalence, even faster in mainland of China increased 21.79% per year from 2013 to 2017 [3–4]. Given China's large population, the number of patients with RA might be predicted to keep increasing, account for about one-fourth of the global RA patient population [4]. Moreover, according to the data of the second nationwide sample survey on disability in China, RA ranked as the second leading cause of disability [5].

Optimal management of RA hinges on early diagnosis and the timely identification of modifiable factors that could arrest or slow disease progression. Efforts are ongoing aimed to develop novel biomarkers for this. Over the past decades, there have been major advances in the treatment of RA. The identification of key cytokines that mediate pro-inflammatory pathway has been proven to be new therapeutic targets [2]. Biologic therapies targeting these pathways have consistently demonstrated efficacy in inducing and sustaining remission, thereby yielding substantial improvements in health-related quality of life [6]. However, all of improvements of therapies were accompanied by the increase in the cost of treatment compared to traditional

disease-modifying antirheumatic drugs (DMARDs) [7]. Therefore, RA imposes a considerable economic burden on society, encompassing not only substantial direct healthcare expenditures but also significant indirect costs arising from lost productivity. Nevertheless, comprehensive and contemporary data on the economic burden of RA in China remain scarce.

Cost of illness (COI) is an estimate of the burden of disease in monetary terms, and highly relevant to policy decision-making. Although numerous COI studies of RA have been published, most of them were conducted in Western countries—most notably the United States, the Netherlands, and Canada—leaving a substantial evidence gap in other regions [7]. In Asia, although annual per-patient costs of RA have been reported from Japan, Hong Kong, Taiwan, and Thailand [8–11], these estimates varied widely, reflecting heterogeneity in socioeconomic contexts, geographic settings, population characteristics, health-care systems, and methodological approaches [7]. To date, the evidences of the costs of RA in mainland of China remain scarce. Xu, et al's study published in 2014 estimated annual average total costs for per RA patient as $3,826 based on 829 patients' data of costs in a cross-sectional study conducted in 2009 [12]. Hu et al's study published in 2017 interviewed only 133 RA patients from two hospitals (on in south China, and the other in north China) in 2013 and estimated the annual average direct costs for RA as $2,410 [13]. Both of them were tertiary hospital-based study with poor representation and small sample size. Cao et al reported the cost per patient related to RA in China in 2017 was $907.78 using the two major databases of health insurance programs in urban China [4]. Compared with the previous two studies, the large number of RA patients and good representation of the national urban population were the obvious features of this study. But it relied on the insurance database in urban China, lacking the data in rural areas, which have different insurance systems [4]. Due to the lack of clinical information, RA diagnosis was estimated by the algorithm, which was not verify the accuracy in Cao et al's study [4]. Considered that Etanercept was the first biological DMARDs included in the National Reimbursement Drug List in 2017, it indicated the changes in healthcare reimbursement policies of RA during the recent years [14]. All of these showed that the existing evidences were difficult to provide a reliable sight into the annual per capita costs of RA patients in China at current.

Chinese registry of rheumatoid arthritis (CREDIT) was established in 2016 supported by the Chinese Rheumatism Data Center (CRDC) aimed to enhance the application of the "treat-to-target (T2T)" strategy nationwide [15]. Up to date, it is the largest resource platform of RA patients in China and provides the important information to understand the "real-word" situation of Chinese RA patients. However, the characteristics of the sample population based on CREDIT might deviate from the characteristics of the overall population due to non-random sampling, none or low response, or loss-to-follow-up, and so on. Inverse probability weighting (IPW), as the one of applications of propensity score techniques initially proposed by Rosenbaum & Rubin, relies on building a statistic model to estimate the probability based on a set of observed covariates for a particular person, then using the predicted probability as a weight in subsequent analyses, which has been used for controlling the confounding or correcting for selection bias caused by none response or loss to follow up in observational studies [16].

In order to obtain a robust estimation of the cost of RA patients in China, we conducted the multi-centers, cross-sectional online cost survey based on CREDIT cohort. Furthermore, we created the weighted patients using IPW in which their demographic and clinical characteristics were closed to the target RA patients of CREDIT cohort. Finally, we estimated annual per capita costs and their influence factors using the Bootstrap method. These findings not only helped to clearly understand on where the costs of RA were incurred in the era of China implementing T2T treatment strategies for rheumatoid arthritis, but also provided the important evidence for further economic decision-making in order to achieve the ultimate goal of improving patients' outcomes.

## Materials and methods

### Study design

This was the multicenters, cross-sectional online study based on the CREDIT cohort in China December 22, 2020, to December 2, 2022. It was conducted in accordance with the Declaration of Helsinki and approved by the Institutional

Review Board of the institute of Basic Medical Sciences, Chinese Academy of Medical Sciences (Project No.063–2020). All participants provided written informed consent. This study was reported in accordance with the STROBE (STrengthening the Reporting of OBservational studies in Epidemiology) guidelines [17]. The detailed information of CREDIT cohort including research design, data collection, follow-up had been published in previous study [15]. The flow chart of this study design was shown in Fig 1.

### Target population and sample patients

The target population included all the RA patients from CREDIT who fulfilled the 2010 American College of Rheumatology (ACR)/European Alliance of Associations for Rheumatology (EULAR) classification criteria [18], and updated the clinical records on CRDC during last 30 days. During the period of study, the total of 18,507 patients from CREDIT updated the clinical records and met the criteria of recruitment.

We send online survey invitation to each of these target populations. When they agreed to participate and signed the informed consent, they completed an online questionnaire that retrospectively collected all outpatient and inpatient expenditures incurred during last year prior to the survey. These respondents constituted the sample of this cross-sectional study (Sample).

Following pilot testing and expert review, questionnaires completed in <3 min or with >50% missing data were invalid. According to this criteria, 12 patients were removed from our analysis. Finally, the total of 1293 patients (about 6.99%) from 152 hospitals in 29 provinces of China signed the informed consent form and completed the valid questionnaire in the online cross-sectional study, which was the sample of convenient sampling according to the willingness to participate.

### Data collection

For all the target population, the patients' complication, clinical characteristics, and regiments of treatment were tracked from the database of the Chinese Rheumatology Information System (CRIS). For the sample of cross-sectional study, the more information was collected by the questionnaire about the expenditures. For the outpatients of RA, the expenditures for the recent outpatient visits were collected, including medications, ancillary services (laboratory tests, radiology tests, or antibody tests), physician charges, transportation, food expenses, as well as days lost from work from patient and their accompanies. The numbers of outpatient visits in the last year was also reported by the patients. For those self-reported inpatients, additional information of inpatient expenditures in last one year was required to be recalled and collected. In this study, outpatient and inpatient expenditures included RA-related care plus all comorbidities, irrespective of their association with RA.

### Measurements

The source of the patients was divided into three geographic areas (eastern, central, and western regions) according to the criteria of the National Bureau of Statistics of China [19]. The disease duration was defined as the years from the onset of RA diagnosis, and classed into three groups (≤1 years, 1–7years, and >7 years) by the quantile. Those patients less than 3 months of RA duration were identified as newly treated patients. Disease activity was divided into remission (DAS28-CRP score < 2.6), low (2.6 ≤ DAS28-CRP score<3.2), moderate (3.2 ≤ DAS28-CRP score≤5.1) and high (DAS28-CRP score >5.1) according to the American College of Rheumatology criteria [20].

### Direct and indirect cost calculation

The annual total costs of RA were calculated using a bottom-up approach (person-based data) and included direct and indirect costs during the last year prior to the survey (See Fig 1). All costs were measured in the Chinese Yuan Renminbi (CNY).

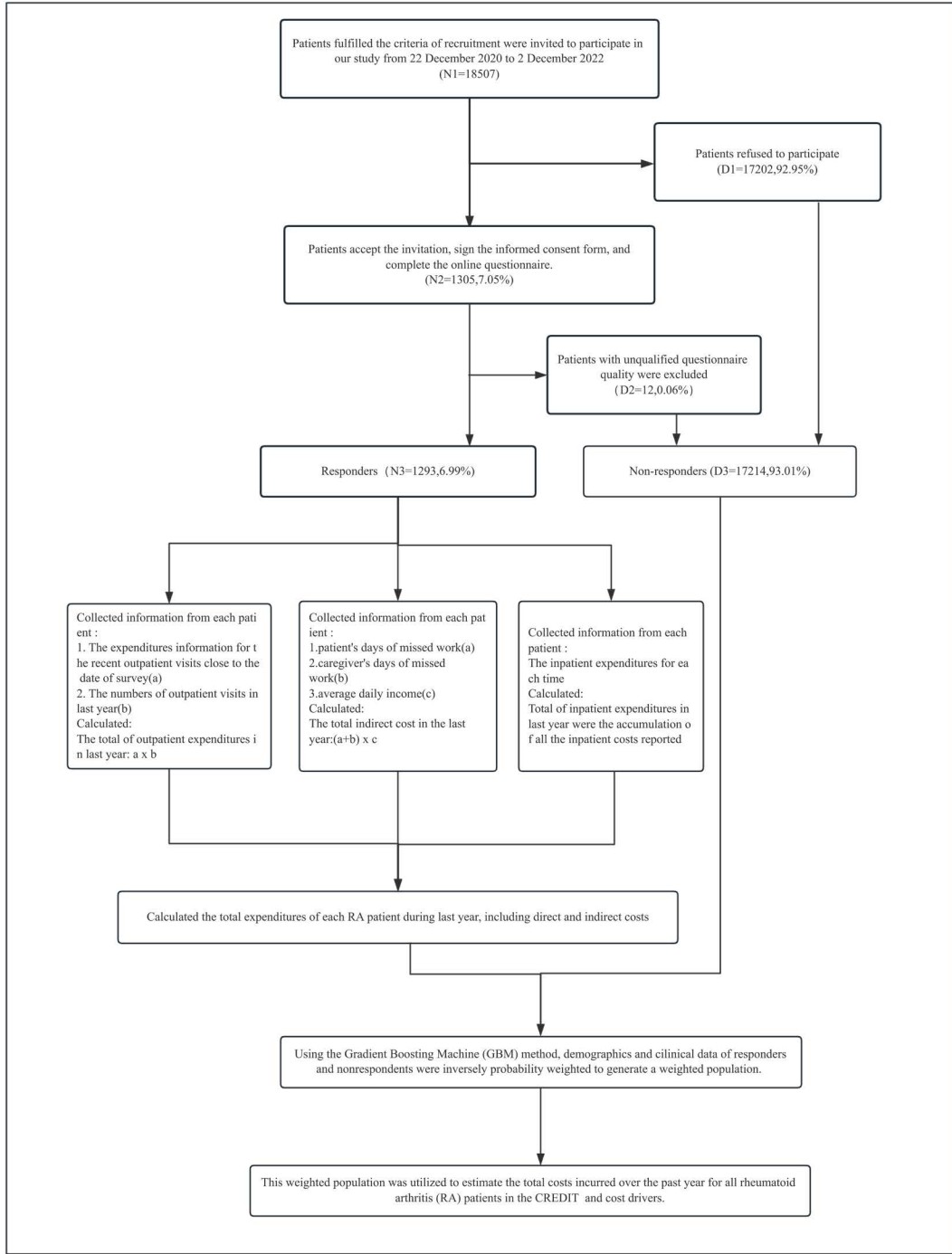

**Fig 1. Flow chart of the study design.**

**Annual direct cost estimates.** For each RA patient, the annual direct costs included the total outpatient and inpatient expenditures. The total outpatient expenditures were calculated by the multiplication of recent costs in outpatient visit and the self-reported numbers of visits during the last year. The inpatient expenditures were the accumulation of all the

inpatient costs reported by patients themselves. For the patients without hospitalization, the inpatient expenditures were recorded as zero. Furthermore, the annual direct costs of RA also were divided into medical and non-medical costs. The former included the charges of physician, drugs, laboratory tests and imaging examinations, antibody examinations, physiotherapy and other services (e.g., purchase of medical aids), hospitalizations, and surgeries. The latter were the costs incurred in the pathway to care and/or to access the services, including transportation, food, and accommodation costs, as well as the informal care costs.

**Indirect cost estimates.** The human capital approach (HCA) was used to estimate the indirect costs. The HCA measures the loss of productivity of a patient or caregiver due to work absenteeism because of outpatient visits or inpatients within the past year. The lost production was evaluated by the opportunity cost of hiring a replacement from the labor market [21]. The formula for computing indirect costs involves multiplying the patient's days of work absenteeism by the patient's average daily income and adding the caregiver's days of missed work multiplied by their average daily income. The average daily income was used daily per capita disposable income of the province where the patient comes from as a proxy measure in this study.

## Statistical analysis

Considered that the extreme value of variables might overestimate the total costs, we used values of 99th percentile to replace in order to reduce their influence. The distribution of the data of costs were typically right-skewed, so the Bootstrap method was used to calculate the arithmetic mean and the Bootstrap 95% confidence interval (CI) in this study [22]. The arithmetic means of costs were calculated with 1000 bootstrap samples by resampling with replacement. The Bootstrap 95%CI for the mean was estimated by empirical bootstrap method [23,24]. The comparisons of costs between groups were also using the Mann–Whitney U test or Kruskal–Wallis's test. A p value of <0.05 was considered statistically significant.

Considered the demographic and clinical differences between the sample in our survey and the target RA patients on CREDIT cohort, the response probability (called "*propensity score*", range from 0 to 1) was estimated for each RA patients who met the criteria of recruitment by Gradient Boosting Machine (GBM) method based on a set of observed covariates (including demographics and clinical features). Compared to logistic regression models usually used to estimate the propensity score, GBM demonstrated better precision in propensity index estimation, with more stable inverse probability weighting values, especially in larger sample sizes [25]. As a tree-based integrated method, GBM could capture non-linearities and high-order interactions automatically, mitigate model misspecification through iterative gradient optimization, and yield smoothly calibrated probabilities that avert extreme propensity scores (0 or 1). This stabilized the inverse-probability weights and minimizes their undue influence on estimates [25]. After then, weights were calculated for each RA patients as 1/propensity score for the RA in our sample, and 1/(1-propensity score) for others [16]. By the IPW, patients with a lower propensity score received larger weights and their influence on the estimation could be increased, so that the difference of characteristics were balance. Therefore, the IPW population, similar demographics and clinical characteristics to the target RA patients who met the criteria of recruitment on CREDIT, were created to estimate the annual costs and investigate the influence factors in our study. The Influence factors of annual total costs were identified using weighted multivariate line regression, in which annual total costs (dependent variable) were normalized by a logarithmic transformation. The stepwise approach was used with a significance level of only 0.05 to include or exclude the variables in the model.

Furthermore, the sensitivity analysis was conducted in this study. On the one hand, we calculated the propensity score by Logistic regression, based on which the weighted annual per capital direct, indirect and total costs were estimated. On the other hand, we took the average per capita Gross domestic product (GDP) of the province where the patient comes from as the proxy measures to estimate the indirect costs.

The statistical analyses were performed using SAS 9.4 software (SAS Institute, Cary, NC, USA) and R software (version 4.1.0) using the 'twang' package.

## Results

### Socio-demographic and clinical characteristics

During the period of our study, the total 18,507 patients from CREDIT updated the clinical records and met the criteria of recruitment. Of them, 1293 RA patients (sample) from 152 hospitals belonged to 29 provinces in China were completed the valid questionnaire. Of them, majority of these patients were females (82.4%), with mean age of 47.73±13.86 years old, median duration of RA 3 years (IQR:1–7 years). 46% of them were in moderate or high disease activity, 32% used biological DMARDs (bDMARDs) or targeted synthetic DMARDs (tsDMARDs), and 17% self-reported hospitalization during last year. The characteristics of weighted population by IPW (IPW population) were more closed to those of RA target patients met the criteria of recruitment during the study period on CREDIT cohort (See Table 1 and more detail in S1 Table, the distribution of propensity score estimated by GBM seen in S1 Fig.). Therefore, the IPW population used to estimate the annual per capita costs of RA patients in this study.

### Direct and indirect cost of RA patients estimated by IPW

The annual per capita direct, indirect and total costs of RA patients estimated by IPW were shown in Table 2. The Bootstrap mean of annual total costs was 41,971 CNY (95%CI: 37,107–47,046 CNY), of them 32,448 CNY estimated for direct costs (Bootstrap 95%CI: 28,412–37,030 CNY), accounting for more than 75%. Moreover, the annual medical costs were estimated for 28,792 CNY, nearly 89% of direct costs. (Table 2).

The detail composition of the direct costs was shown as Fig 2 (more details shown on S2 Tables). For the direct medical costs, the drug expense accounted for the greatest proportion (59.7%), followed by the expense for laboratory test, imaging or antibody examination (22.7%). For the direct non-medical costs, the proportion of transportation expense was 35.6%, follow by food expense (27.3%) and accommodation expense (19.4%).

In sensitive analysis, when taken average daily per capita GDP of province where patients came from as the proxy of daily incomes the average annual total costs were estimated 54,097 CNY (95%CI: 48,364–60,552 CNY), in which 60% were direct costs, and medical costs accounted for 88.7% of direct costs (S3 Table). Using Logistic regression model to calculate the propensity score, the weighted Bootstrap mean of total costs were 44,315 CNY (95%CI: 39,113–50,410 CNY), in which 77% were direct costs, and medical costs accounted for 89% of direct costs. Similar proportion of costs were found both GBM method and Logistic method (S4 Table).

### Influence factors related to costs

According to Table 3, geographical regions, medical insurance, hospitalization during the last year, history of joint replacement, moderate or high active disease activity status, as well as treatment of RA, might associated with the average annual costs of RA patients.

Furthermore, the results of weighted stepwise multivariable regression model indicated that the total costs of RA were related to geographical regions, hospitalized during last year, history of comorbidity (including joint replacement, allergy disease), disease activity, and treatments (including bDMARDs or tsDMARDs, glucocorticoid) when controlling for the covariates. (See Table 4)

For the patients who hospitalized during last year, compared to those of non-hospitalized, there was an increment of 228% (amount: 26,315 CNY, p<0.01) in annual total cost controlling for covariates. The total costs also increased nearly 78.3% or more among those with joint replacement (amount: 9,038 CNY, p=0.02) or allergy disease (amount: 3,391 CNY p=0.04). The disease activity was also important factors to influence on the total cost, compared to remission, low,

**Table 1. RA patients' demographics and clinical characteristics.**

| Characteristics | RA target patients during the study period on CREDIT | Sample | IPW Population |
|---|---|---|---|
| | (N = 18507) | (N = 1293) | (N = 18465) |
| **Age, mean (SD)** | 52.66(13.05) | 47.73(13.86) | 52.69 (13.07) |
| **Gender, n(%)** | | | |
| Male | 3290(17.8) | 228(17.6) | 3276 (17.7) |
| Female | 15217(82.2) | 1065(82.4) | 15188 (82.3) |
| **Geographical regions, n(%)** | | | |
| Western | 4790(25.9) | 271(21.0) | 4797 (26.0) |
| Central | 6022(32.5) | 359(27.8) | 6009 (32.5) |
| Eastern | 7695(41.6) | 663(51.3) | 7659 (41.5) |
| **Medical insurance, n(%)** | | | |
| No | 6008(32.5) | 230 (17.8) | 6058 (32.8) |
| Yes | 12499(67.5) | 1063 (82.2) | 12439 (67.2) |
| **Duration (years), median [p25, p75]** | 3.00 [0.00, 8.00] | 3.00 [1.00, 7.00] | 3.00 [0.00, 8.00] |
| **Newly treated patients, n(%)** | 4316(23.3) | 259(20.0) | 4345 (23.5) |
| **History of comorbidity, n(%)** | | | |
| Fragility fracture | 231(1.2) | 20(1.5) | 224 (1.2) |
| Joint replacement | 291(1.6) | 21(1.6) | 286 (1.5) |
| Neoplasms | 191(1.0) | 19(1.5) | 186 (1.0) |
| Allergy disease | 1089(5.9) | 99(7.7) | 1077 (5.8) |
| Diabetes | 937(5.1) | 43(3.3) | 941(5.1) |
| Hypertension | 2884(15.6) | 156(12.1) | 2887 (15.6) |
| Hyperlipidemia | 739(4.0) | 70(5.4) | 730 (4.0) |
| **Family history of RA, n(%)** | 713(3.9) | 90(7.0) | 704 (3.8) |
| **DAS28-CRP, mean (SD)** | 3.65(1.55) | 3.27(1.51) | 3.66 (1.55) |
| **RF, median [IQR]** | 92.40 [31.00, 236.00] | 91.00 [32.90, 200.00] | 91.96 [31.00, 236.00] |
| **ESR, median [IQR]mm/h** | 23.00 [12.00, 45.00] | 20.00 [10.00, 36.00] | 23.00 [12.00, 45.00] |
| **CRP, median [IQR]mg/dl** | 5.43 [1.94, 16.03] | 3.48 [1.30, 10.35] | 5.60 [2.00, 16.30] |
| **Patient pain VAS score, median [IQR]** | 3.80 [2.20, 5.30] | 3.20 [1.70, 5.10] | 3.80 [2.20, 5.30] |
| **Global disease VAS score (patient), median [IQR]** | 3.80 [2.20, 5.40] | 3.30 [2.00, 5.10] | 3.80 [2.20, 5.40] |
| **Global disease VAS score (physician), median [IQR]** | 3.80 [2.10, 5.40] | 3.10 [1.80, 5.10] | 3.80 [2.20, 5.40] |
| **Disease activity, n(%)** | | | |
| Remission | 5173(29.0) | 491(39.5) | 5138 (28.9) |
| Low | 2544(14.3) | 180(14.5) | 2541(14.3) |
| Moderate | 6761(37.9) | 405(32.6) | 6757 (38.0) |
| High | 3363(18.8) | 167(13.4) | 3365(18.9) |
| **Treatment, n(%)** | | | |
| csDMARDs | 11701(63.2) | 1049(81.1) | 11651 (63.1) |
| NSAIDs | 2532(13.7) | 144(11.1) | 2541(13.8) |
| bDMARDs/tsDMARDs | 4505(24.3) | 417(32.3) | 4120(22.3) |
| Glucocorticoid | 4129(22.3) | 284(22.0) | 4478 (24.3) |
| Drug for osteoporosis | 3921(21.2) | 223(17.2) | 3931 (21.3) |

Note: csDMARDs: Conventional Synthetic Disease-Modifying Anti-Rheumatic Drugs

NSAIDs: Non-Steroidal Anti-Inflammatory Drugs

bDMARDs/ tsDMARDs: Biological Disease-Modifying Anti-Rheumatic Drugs or Targeted Synthetic Disease-Modifying Anti-Rheumatic Drugs

Table 2. **Average annual costs estimated by Bootstrap method among IPW population (Unit: CNY).**

| | Weighted Median (IQR) | Weighted Mean±SD | Weighted Bootstrap Mean (95%CI) |
|---|---|---|---|
| **Direct costs** | 12,200 (5,682−32,601) | 32,375±82,197 | 32,448(28,412−37,030) |
| *Medical* | *9,986 (4,397− 28,139)* | *28,719±78,161* | *28,792(24,893−33,208)* |
| *Non-Medical* | *1,200 (380− 3,750)* | *3,657±9,984* | *3,656(3,130−4,251)* |
| **Indirect costs** | 1,488 (0-6,736) | 9,520±21,488 | 9,523(8,343−10,726) |
| **Total costs** | 17,272 (7,582− 48,752) | 41,895±91,238 | 41,971(37,107−47,046) |

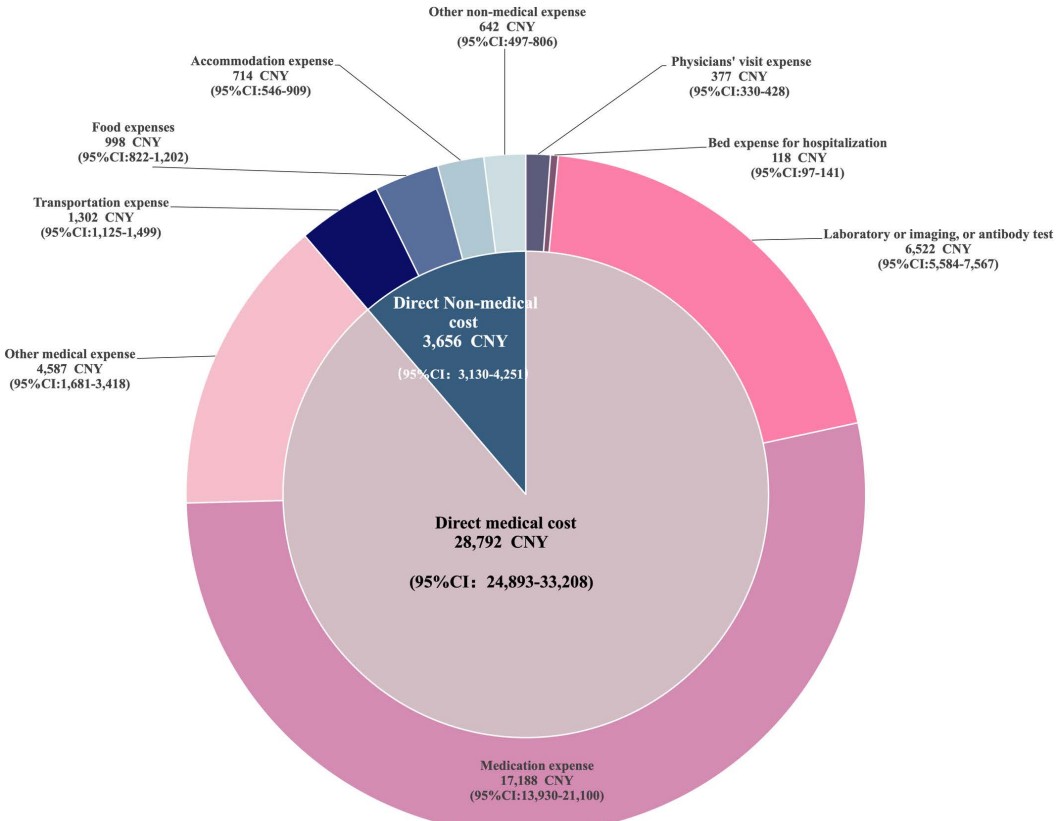

**Fig 2. The detail composition of direct medical cost and non-medical cost of RA patients in China.**

moderate, and high state of disease could increase 35%, 27%, and 49%, respectively. Moreover, the total costs increased significantly (61%, amount: 6,998 CNY, p<0.01) for use of biologics, and increased 30% (amount: 3,472 CNY, p=0.02) for use of glucocorticoids.

## Discussion

To the best of our knowledge, we firstly produced the weighted pseudo-population by the IPW method, in which the demographic and clinical characteristics were more similar to those of the target RA patients on CREDIT cohort. Moreover, we used the Bootstrap approach to estimate the mean and uncertainty in the annual costs of RA considered their typically

**Table 3. Details of average annual total costs estimated by Bootstrap method among IPW population. (Unit: CNY).**

| | Bootstrap Mean (95%CI) | p value |
|---|---|---|
| **Age groups** | | 0.454 |
| 18–60 years | 42773(35083-46769) | |
| ≥60 years | 43985(37074-55283) | |
| **Gender** | | 0.515 |
| Male | 40957(37119-56577) | |
| Female | 43455(35681-47233) | |
| **Geographical regions** | | <0.001* |
| Western | 61728(39470-72321) | |
| Central | 31315(25284-36015) | |
| Eastern | 41701(39025-52046) | |
| **Medical insurance** | | 0.045 |
| No | 35414(26109-40924) | |
| Yes | 44659(39390-51559) | |
| **Hospitalization during last year** | | <0.001 |
| No | 23695(20152-27452) | |
| Yes | 79644(58789-107102) | |
| **Newly treated patients** | | 0.695 |
| No | 41967(36358-52676) | |
| Yes | 43277(35847-47715) | |
| **History of Comorbidity** | | |
| *Fragility fracture* | | 0.070 |
| None | 43131(37631-47352) | |
| Yes | 35613(17019-42541) | |
| *Joint replacement* | | 0.016 |
| None | 42239(36061-46562) | |
| Yes | 90017(63166-156058) | |
| *Neoplasms* | | 0.703 |
| None | 42681(37009-46758) | |
| Yes | 65380(17169-95650) | |
| *Allergy disease* | | 0.507 |
| None | 42194(36260-47313) | |
| Yes | 52908(33834-59224) | |
| *Diabetes* | | 0.479 |
| None | 42937(37553-47599) | |
| Yes | 45273(20721-57081) | |
| *Hypertension* | | 0.349 |
| None | 41634(35812-47211) | |
| Yes | 53076(36971-60760) | |
| *Hyperlipidemia* | | 0.824 |
| None | 42886(37057-47191) | |
| Yes | 45269(30310-59115) | |
| **Family history of RA** | | 0.414 |
| None | 43448(37378-48261) | |
| Yes | 37223(22126-51422) | |
| **Disease activity** | | <0.001 |

*(Continued)*

**Table 3.** (Continued)

| | Bootstrap Mean (95%CI) | p value |
|---|---|---|
| **Remission** | 28566(23681-33194) | |
| Low | 49623(35779-59320) | |
| Moderate | 43994(35316-52014) | |
| High | 79798(44679-90641) | |
| **Treatment of RA** | | |
| ***csDMARDs*** | | 0.732 |
| None | 44150(31882-49864) | |
| Yes | 42751(37234-49051) | |
| ***NSAIDs*** | | 0.049 |
| None | 43888(38082-49723) | |
| Yes | 36045(24397-42249) | |
| ***bDMARDs/tsDMARDs*** | | 0.005 |
| None | 37436(32304-44150) | |
| Yes | 54733(46024-63566) | |
| ***Glucocorticoid*** | | 0.256 |
| None | 40376(35149-46330) | |
| Yes | 52390(38415-55625) | |
| ***Drugs for Osteoporosis*** | | 0.410 |
| None | 41148(35629-46190) | |
| Yes | 51974(35295-62006) | |

Note: csDMARDs: Conventional Synthetic Disease-Modifying Anti-Rheumatic Drugs

NSAIDs: Non-Steroidal Anti-Inflammatory Drugs

bDMARDs/ tsDMARDs: Biological Disease-Modifying Anti-Rheumatic Drugs or Targeted Synthetic Disease-Modifying Anti-Rheumatic Drugs

right-skewed distribution. The average annual total costs per patient was 41,971 CNY (95%CI: 37,107–47,046 CNY) in our study, in which the direct costs accounted for 75% or over. The medical costs also estimated 28,792 CNY (95%CI: 24,893–33,208 CNY), nearly 89% of direct costs, and even half of them was medication expense. The disease activities, hospitalization, history of comorbidity (e.g., joint replacement or allergy disease), the treatment of biologics or glucocorticoids were found to substantially increase the total costs of RA in China in our study. There also were the significant differences in the geographical distribution of the total cost and the western region were higher than that in the central and eastern regions. All of these results provided a reliable sight into economic burden of RA for the individuals in the real-world clinical practice context of the era of implementing treat-to-targe strategies in China.

To date, the evidences of the costs of RA patients remained insufficient in mainland of China. Compared to the total costs of per RA patient as $3,826 published by Xu et al. in 2014 [12] and $2,410 (RMB) published by Hu et al. in 2017 [13], our estimation of 41,971 CNY (or $6,507 based on the 2021 RMB to US dollar exchange rate) of total cost per patient were increased by 0.7 times or 1.7 times respectively. On the one hand, the most important changes in RA treatment were reflected in the treatment strategy in the recent decade, advocating early T2T approach as a guiding principle, which entails intensive treatment and regular follow-up with the goal of achieving low disease activity or clinical remission [26–28]. And advanced approaches emphasize on the importance of attaining at least 50% improvement in disease activity within 3 months of drug administration [29]. On the other hand, with the introduction of tumor necrosis factor (TNF) inhibitors, followed thereafter by application of interleukin-6 receptor (IL-6R) inhibitors, T-lymphocyte co-stimulation inhibitors (Abatacept), B cell depletion (Rituximab), as well as Janus Kinase (JAK) inhibitors, the treatment of RA in China

**Table 4. Results of the weighted linear regression models (dependent variable: log-transformed average annual total costs).**

| | Weighted Multiple Model (Method: Full)) | | | Weighted Multiple Model (Method: Stepwise) | | | % of Increments or Decrements (Stepwise) | |
|---|---|---|---|---|---|---|---|---|
| | β | SE | p value | β | SE | p value | %ᵃ | Amount (CNY)ᵇ |
| **Intercept** | 9.639 | 0.275 | <0.001 | 9.353 | 0.128 | <0.001 | N.A | 11,537 |
| Age | −0.005 | 0.003 | 0.087 | | | | | |
| Gender(ref:Male) | −0.058 | 0.091 | 0.525 | | | | | |
| Geographical regions (ref: West) | | | | | | | | |
| Central | −0.377 | 0.101 | <0.001 | −0.318 | 0.134 | 0.018 | −27.3 | −3,147 |
| East | −0.265 | 0.091 | 0.004 | −0.245 | 0.119 | 0.039 | −21.7 | −2,507 |
| Medical insurance (ref: None) | −0.077 | 0.09 | 0.393 | | | | | |
| Newly treated patients (ref: No) | 0.041 | 0.091 | 0.649 | | | | | |
| Hospitalization during last year (ref: None) | 1.181 | 0.094 | <0.001 | 1.188 | 0.101 | <0.001 | 228.1 | 26,315 |
| **History of comorbidity** | | | | | | | | |
| *Fragility fracture(ref: None)* | −0.014 | 0.275 | 0.96 | | | | | |
| *Joint replacement(ref: None)* | 0.302 | 0.269 | 0.261 | 0.578 | 0.248 | 0.02 | 78.3 | 9,038 |
| *Neoplasms(ref: None)* | 0.325 | 0.278 | 0.243 | | | | | |
| *Allergy disease (ref: None)* | 0.19 | 0.129 | 0.139 | 0.258 | 0.125 | 0.039 | 29.4 | 3,391 |
| *Diabetes(ref: None)* | −0.098 | 0.2 | 0.623 | | | | | |
| *Hypertension(ref: None)* | 0.203 | 0.114 | 0.075 | | | | | |
| *Hyperlipidemia(ref: None)* | 0.144 | 0.16 | 0.368 | | | | | |
| Family history(ref: None) | 0.096 | 0.135 | 0.479 | | | | | |
| Disease activity(ref: Remission) | | | | | | | | |
| Low | 0.248 | 0.105 | 0.018 | 0.303 | 0.144 | 0.036 | 35.4 | 4,085 |
| Moderate | 0.179 | 0.084 | 0.033 | 0.235 | 0.115 | 0.041 | 26.5 | 3,052 |
| High | 0.362 | 0.111 | 0.001 | 0.402 | 0.207 | 0.053 | 49.4 | 5,701 |
| **Treatment** | | | | | | | | |
| *csDMARDs(ref: None)* | 0.087 | 0.091 | 0.337 | | | | | |
| *NSAIDs (ref: None)* | −0.077 | 0.111 | 0.491 | | | | | |
| *bDMARDs/tsDMARDs (ref: None)* | 0.515 | 0.077 | <0.001 | 0.474 | 0.094 | <0.001 | 60.7 | 6,998 |
| *Glucocorticoid (ref: None)* | 0.301 | 0.086 | <0.001 | 0.263 | 0.108 | 0.015 | 30.1 | 3,472 |
| *Drug for osteoporosis (ref: None)* | 0.15 | 0.094 | 0.11 | | | | | |

Note: ᵃCalculated as: (exp(coefficient) − 1) × 100;

ᵇCalculated as: (Exp(coefficient) − 1) × exp (Intercept);

csDMARDs: Conventional Synthetic Disease-Modifying Anti-Rheumatic Drugs

NSAIDs: Non-Steroidal Anti-Inflammatory Drugs

bDMARDs/ tsDMARDs: Biological Disease-Modifying Anti-Rheumatic Drugs or Targeted Synthetic Disease-Modifying Anti-Rheumatic Drugs

has gradually begun to enter the targeted era [30]. In our study, nearly one-quarter of the patients receiving biologic or target synthetic therapies, which was only 3% reported by Hu et al [13]. All of these reduced the comorbidity and mortality of RA and help more RA patients to achieve the drug-maintained remission. But they also significantly increased the costs related to treatment. In addition, the majority of total costs were direct cost, in which more than 50% for medication expense, which was consistent with the previously studies in China [12,13]. According to the data of the National Bureau of Statistics of China, the annual per capita disposable income was 36,883 CNY in 2022 [19]. It indicated that the annual

per patient's direct costs (32,448 CNY) estimated in our study accounted for approximately 88% of per capita disposable income. Moreover, only 17% of RA patients reported the hospitalization during the last year in our study, so the outpatient expenditure was main composition of cost. Although nearly 67.5% of patients self-reported to have medical insurance, the outpatient expense was reimbursed in a very low proportion slightly varied by provinces in China [31]. Cao et al. published in 2025 reported the $907.78 per-capita cost of RA patients based on database of health insurance in urban of China in 2013–2017 years. Due to the fact that the patient's out-of-pocket expenses and patients uncovered in medical insurance were not considered, it might underestimate the costs of RA in real-word of mainland of China [4]. Overall, the results in our study implied most of patients had to pay the expenses from out-of-the pocket by themselves or their family, which induce to have to face financial hardship due to RA with long-term duration and incurable features, especially for those living in close to or below the poverty line families with sever disease and/or comorbities.

Biological and targeted synthetic DMARDs were revolutionary in RA treatment and reconfigured cost compositions in biologic era [32]. Taking USA as example, the first biologic (etanercept) was introduced in 1998, and then widely adopted by RA patients, which has doubled the proportion of direct costs in the total cost between 1978 and 2002 reported in the systematic review conducted by Rat and Boissier [33]. In China, nearly 25% of RA patients were treated by bDMARDs or tsDMARDs in our study, which was far lower than the usage rate of drug in Europe or North American [33]. On the one hand, it indicated that the costs of RA might continue to rise with advocating T2T approach in future of biologic era. On the other hand, it was undeniable that economic factors might be the main resistance to the widespread use of targeted therapies considered the more expensive costs. The improvement or reform of the medical insurance system may be an effective measure to break through this dilemma in China. In addition, the disease activity and comorbidity were found to significantly increase the total costs of RA in our study, which indicated that the early and intensive treatment strategies for RA patients might be more cost-effective clinical decision to improve the clinical outcomes and quality of life. Compared to those of RA patients from central or eastern regions, the more costs among the patients from the western regions were founded in our study. It might be explained partly that the proportion of moderate and severe patients among RA patients from western regions was higher due to the limitation of underdeveloped economic and the low insurance coverage rates.

Based on the cost per patient estimated in our study, given the vast of RA patients and the trend of increasement of RA prevalence in China, the policy decision-makers and researchers need to recognize the increasingly severe economic burden associated with RA, which is progressively escalating and presenting a significant challenge to our society. To tackle these challenges, future research need focus on the following key areas: First, the sustained implementation of treat-to-target strategies for RA patients and the promotion of regional equity in treatment access are vital, as they might effectively reduce the economic burden of RA. Second, accelerating pharmacoeconomic studies is imperative. By leveraging market competition to lower drug prices, especially for biologics and targeted therapies, and through strategic negotiations with pharmaceutical companies, it is possible to expand the list of reimbursable RA medications and significantly reduce out-of-pocket costs for patients. These measures, combined with targeted adjustments to health insurance policies, can effectively mitigate the economic burden on individuals and families.

However, there were still limitations in our study. Firstly, the indirect costs for patients estimated using the human capital approach, but not included the costs of premature retirement by RA and intangible costs, that might underestimate the indirect costs. Secondly, RA patients frequently carried multiple comorbidities, not all of which were RA-related. Because we captured their total annual healthcare spending, our per-patient costs might overestimate costs attributable solely to RA. Thirdly, the estimation of costs was based on data collected by patient-self reported and there might be recall bias. In order to improve the accuracy as much as possible, we only invited the patients whose clinical records were updated by the clinicians who they visited during last 30 days, and participants were requested to recall the expense information of this recent visit in detail and frequency of visits last year. Moreover, at the initial stage, 53 patients were randomly selected to collect information repeatedly in two weeks to assess agreement of information. Fourthly, given the cross-sectional design, we couldn't exclude the possibility of reverse causation or residual confounding. Therefore, the observed

associations between the influence factors and expenditures of RA warrant confirmation in further prospective cohort studies. Lastly, considered the overall response rate of only 7%, our cross-sectional study was susceptible to non-response bias. Although IPW mitigated imbalances in observed characteristics, residual confounding from unmeasured factors (such as income, health-seeking behaviors, et al.) might persist, potentially limiting the external validity of our findings to the wider RA population.

## Conclusions

In summary, this study provided reliable insight into evaluating the economic burden of RA for the individuals and their families in the real-world clinical practice of China. The T2T approach might decreased the costs of RA by effectively improve clinical outcomes. The biological and targeted synthetic DMARDs brought the light to the treatment of RA patients, while they increased the cost of RA treatment. Therefore, further economic evaluations of new strategies in RA treatment are needed to assist clinicians and decision-makers in making informed choices.

## Supporting information

**S1 Fig. The distribution of propensity score estimated by GBM for sample in online survey and none-sample on CREDIT cohort.**
(TIF)

**S1 Table. All the patients' demographics and clinical characteristics and comparison between sample and none-sample, weighted sample and weighted non-sample.**
(DOCX)

**S2 Table. Annual per capital costs estimated by the IPW pseudo-population RA patients (Unit: CNY).**
(DOCX)

**S3 Table. Average Annual costs estimated taken average per capital GDP as the proxy by the IPW population of RA patients in China (Unit: CNY).**
(DOCX)

**S4 Table. Average Annual costs estimated by the IPW (logistic regression model) population of RA patients in China (Unit: CNY).**
(DOCX)

## Acknowledgments

The authors would like to thank all the patients who participated in this survey. We also thanked all CREDIT centers for clinical data collection and Health Cloud as the system provider.

## Author contributions

**Conceptualization:** Bing Yu, Lin Qiao, Qian Wang, Mengtao Li, Yanhong Wang, Xinping Tian.

**Data curation:** Nan Jiang, Qian Wang.

**Formal analysis:** Bing Yu, Lu Li, Yanhong Wang.

**Funding acquisition:** Mengtao Li, Xinping Tian.

**Investigation:** Lin Qiao, Li Wang, Nan Jiang, Qian Wang, Yanhong Wang.

**Methodology:** Lu Li, Keying Zuo, Liangming Li, Li Wang, Yanhong Wang, Xinping Tian.

**Supervision:** Li Wang, Qian Wang, Mengtao Li, Yanhong Wang, Xinping Tian.

**Validation:** Keying Zuo, Liangming Li.

**Visualization:** Yanhong Wang.

**Writing – original draft:** Bing Yu, Lin Qiao, Yanhong Wang, Xinping Tian.

**Writing – review & editing:** Bing Yu, Lin Qiao, Lu Li, Mengtao Li, Yanhong Wang, Xinping Tian.

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
