## [Decision Letter · Decision Letter 0]

2 Jun 2025

Dear Dr. Wang,

Thank you for submitting your manuscript to PLOS ONE. After careful consideration, we feel that it has merit but does not fully meet PLOS ONE’s publication criteria as it currently stands. Therefore, we invite you to submit a revised version of the manuscript that addresses the points raised during the review process.

**ACADEMIC EDITOR: **

**Title and abstract**

- The abstract should include the total number of patients.

**Introduction**

- You may highlight the epidemiology of RA, including its prevalence and female-to-male ratio.DOI: 10.1177/03000605231204477

- You may draw attention to the RA patient's annual healthcare costs. DOI: 10.3389/fmed.2023.1221393

- The gaps in knowledge and rationale for the study need to be mentioned.

**Patients and methods**

- Please outline this section following the STROBE guidelines.

- Describe the setting, locations, exposure, follow-up, and data collection

-  How were the patients selected (e.g., consecutively, randomly, or selectively)?

- You need to state in the Methods section that you have followed STROBE guidelines: ‘The reporting of this study conforms to STROBE. (Insert new reference number)

**Results**

- Abbreviations should be explained as subtitles below the Figures / Tables

**Discussion**

- A comparison of your results and the relevant previous studies should be made.

- How can future research build on these observations? What are the key experiments that must be done? 

We look forward to receiving your revised manuscript.

Kind regards,

Wesam Gouda, MD,PhD

Academic Editor

PLOS ONE

Journal Requirements:

“This study was supported by the Chinese National Key Technology R&D Program, Ministry of Science and Technology (2022YFC2504600, 2022YFC3601800), CAMS Innovation Fund for Medical Sciences (CIFMS) (2021-I2M-1-005, 2022-I2M-1-004, 2023-I2M-2-005), The Non-profit Central Research Institute Fund of Chinese Academy of Medical Sciences (2021-PT320-002, 2019-PT330-004), National High Level Hospital Clinical Research Funding (2022-PUMCH-B-013), The Special Science Research for Health Development in Capital (No.2024-1G-2082)

3. In this instance it seems there may be acceptable restrictions in place that prevent the public sharing of your minimal data. However, in line with our goal of ensuring long-term data availability to all interested researchers, PLOS’ Data Policy states that authors cannot be the sole named individuals responsible for ensuring data access (http://journals.plos.org/plosone/s/data-availability#loc-acceptable-data-sharing-methods).

5.  We note that Figures S2 in your submission contain [map/satellite] images which may be copyrighted. All PLOS content is published under the Creative Commons Attribution License (CC BY 4.0), which means that the manuscript, images, and Supporting Information files will be freely available online, and any third party is permitted to access, download, copy, distribute, and use these materials in any way, even commercially, with proper attribution. For these reasons, we cannot publish previously copyrighted maps or satellite images created using proprietary data, such as Google software (Google Maps, Street View, and Earth). For more information, see our copyright guidelines: http://journals.plos.org/plosone/s/licenses-and-copyright.

a. You may seek permission from the original copyright holder of Figures S2 to publish the content specifically under the CC BY 4.0 license. 

Reviewers' comments:

Reviewer's Responses to Questions

**Comments to the Author**

1. Is the manuscript technically sound, and do the data support the conclusions?

Reviewer #1: Yes

Reviewer #2: Yes

Reviewer #3: Yes

2. Has the statistical analysis been performed appropriately and rigorously?

Reviewer #1: Yes

Reviewer #2: Yes

Reviewer #3: Yes

3. Have the authors made all data underlying the findings in their manuscript fully available?

Reviewer #1: Yes

Reviewer #2: Yes

Reviewer #3: Yes

4. Is the manuscript presented in an intelligible fashion and written in standard English?

Reviewer #1: Yes

Reviewer #2: Yes

Reviewer #3: Yes

Reviewer #1: This manuscript addresses an important topic, providing valuable insights into the economic burden of rheumatoid arthritis (RA). The study design and methodological rigor are commendable, and the detailed cost analysis is a notable strength. However, some areas require further refinement to enhance clarity and impact.

Study Design and Methods: The recruitment flowchart is helpful, but the criteria for excluding "unqualified questionnaire quality" (n=22) need clarification. Additionally, the rationale for selecting the Gradient Boosting Machine (GBM) method over other approaches should be elaborated.

Data Presentation: The cost distribution chart is effective, but the discussion on medication expenses could explore potential cost-containment strategies, such as the role of generics or insurance reforms. Consider condensing the frequency tables into supplementary materials for improved readability.

Ethical and Research Transparency: Describe in detail, if applicable, IRB approval, funding, and detailed information about data availability. Include a data availability statement to foster reproducibility.

Interpretation and Recommendations: Even though the study has identified a significant cost component, the discussion should be further extended to cover policy implications in particular, which may include indirect cost reduction strategies such as telemedicine or workplace accommodations. Future research ideas could include extending this framework to other conditions.

In summary, this manuscript is a strong contribution to understanding the economic burden of RA. Addressing the above points will improve the study's clarity, robustness, and practical implications.

Reviewer #2: Reviewer Comments (Minor Revisions):

The manuscript presents a well-designed cross-sectional cost-of-illness study using data from a large national RA registry. The application of inverse probability weighting (IPW) via generalized boosted modeling and the use of bootstrap methods for cost estimation are appropriate and robust. The findings offer valuable insight into the economic burden of RA in China and identify key cost drivers that have clinical and policy relevance.

However, I would suggest two minor revisions to improve the clarity and interpretability of the study:

1. Survey Response Rate and Generalizability: The response rate of approximately 7% (1,293 of 18,507 eligible patients) raises potential concerns about non-response bias. Although IPW adjustment mitigates this to some extent, residual bias from unmeasured confounders (e.g., income, health-seeking behavior) may remain. I recommend briefly expanding the discussion on this limitation and its implications for the generalizability of the findings.

2. Attribution of Costs to RA vs. Comorbidities: It is not entirely clear whether the estimated costs are specific to RA-related healthcare utilization or may include expenditures related to comorbidities. Since RA patients often have overlapping medical conditions, further clarification in the Methods and Discussion sections regarding the attribution of costs would enhance transparency.

These revisions are relatively minor and do not detract from the overall quality of the work. I support publication pending minor revisions.

Reviewer #3: The research entitled "Average Annual Costs of Rheumatoid Arthritis Estimated by Inverse Probability Weighting and Their Predictors: A Cross-Sectional Study Based on the Chinese Registry of Rheumatoid Arthritis (CREDIT) Cohort" explicitly acknowledges a number of caveats and limitations.

1. Cross-Sectional Design: The study employs a cross-sectional approach, indicating that data were gathered at a singular moment in time. This methodological framework constrains the capacity to ascertain causal relationships between predictors, such as disease activity or treatment modalities, and annual expenditures.

2. Data Acquisition and Representational Validity: Data pertaining to outpatient and inpatient expenditures were gathered through online questionnaires, a method that may potentially lead to recall bias or inaccuracies in reporting if patients fail to accurately remember or disclose their expenses.

3. The sample was derived from the CREDIT cohort, and while inverse probability weighting (IPW) was employed to construct a weighted population that mirrors the larger RA patient demographic, there exists a possibility that the sample may not comprehensively represent all RA patients in China, particularly those individuals not included in the registry. The low response rate may still affect representativeness and external validity.

4. Methods of Estimation: Indirect costs were assessed through the human capital approach; however, this method may overlook certain societal expenses, including intangible costs associated with quality of life and unpaid labor. This may lead to an underestimation of the true indirect cost burden in RA.

5. Attribution Challenge in Health Expenditures: The research recognized the challenges associated with differentiating healthcare expenses that can be directly linked to RA from those arising from concurrent comorbid conditions. Consequently, the analysis encompassed all healthcare expenditures reported by patients with RA over the preceding year, which may lead to an overestimation of costs specifically associated with RA.

6. Employing bootstrap methods for cost estimation yields strong confidence intervals; nevertheless, it is contingent upon the premise that the sample is representative and that the resampling effectively reflects the variability within the population.

7. Possible Confounding Variables: Despite the application of multivariate regression and inverse probability weighting to account for confounding factors, there remains the possibility of residual confounding arising from variables that are either unmeasured or inaccurately measured.

8. Generalizability: The research is predicated on data sourced from China, where healthcare systems, cost frameworks, and treatment modalities may exhibit significant variations in comparison to other nations. This disparity potentially constrains the applicability of the findings beyond the specific context of China.

9. Language and Grammar: The manuscript contains several grammatical errors and awkward phrasing.

**Do you want your identity to be public for this peer review?** For information about this choice, including consent withdrawal, please see our Privacy Policy

Reviewer #1: **Yes: ** Kola Adegoke

Reviewer #2: No

Reviewer #3: No

---

## [Author Response · Author response to Decision Letter 1]

15 Jul 2025

ACADEMIC EDITOR:

Title and abstract

- The abstract should include the total number of patients.

Response: Thank you for your suggestion. The information of “In this study, a total of 18,507 patients from the CREDIT database met the recruitment criteria. Among them, 1,293 patients from 152 hospitals across 29 provinces in China completed the questionnaire and were included in our analysis.” has been added in revised manuscript (Abstract, Page2, Line 39-42).

Introduction

- You may highlight the epidemiology of RA, including its prevalence and female-to-male ratio. DOI: 10.1177/03000605231204477

Response: Thank you for your good suggestion, the information of “In China, the prevalence of rheumatoid arthritis aligns closely with global estimates, whereas its incidence appears modestly higher. [3-4]. The most recent large-scale epidemiological study reported an overall prevalence of 368.11 per 100,000 people and an incidence rate of 140.64 per 100,000 person-years in 2017 of China, with a female-to-male ratio of 1.58:1 [4]. During the past decades, the burden of RA has been on a significant upward trend around the world with 7.4% increase in the global age-standardized prevalence, even faster in mainland of China increased 21.79% per year from 2013 to 2017 [3-4]. Given China’s large population, the number of patients with RA might be predicted to keep increasing, account for about one-fourth of the global RA patient population [4].” were added in the introduction. We also revised wording for clarity and accuracy. (Introduction, Paragraph 1, Line 59-68)

- You may draw attention to the RA patient's annual healthcare costs. DOI: 10.3389/fmed.2023.1221393

Response: Thank you for your good suggestion, in the revised manuscript, we updated the introduction to underscore the growing public-health importance of addressing RA costs amid rising prevalence. Moreover, we incorporated the most recent evidence published in 2025 to provide a comprehensive picture of RA-related costs in mainland China; all additions are highlighted in the revised manuscript. (Introduction, Paragraph 1- Paragraph 2)

- The gaps in knowledge and rationale for the study need to be mentioned.

Response: Thank you for the helpful suggestion. We have now identified three published studies on RA costs in Chinese patients; in the revised manuscript we summarize their limitations and emphasize the contribution of our work. (Introduction, Paragraph 3, Line 136-Line 147)

Patients and methods

- Please outline this section following the STROBE guidelines.

Response: Thank you for your suggestion, we revised the ' Materials and Methods' section to follow the STROBE guidelines with clear subheadings and structured reporting. (Materials and Methods, Paragraph1, Line187-189)

- Describe the setting, locations, exposure, follow-up, and data collection

Response: Thank you for your suggestion. Details of the CREDIT registry have been reported previously; the relevant citation has now been added (Materials and Methods, Paragraph1, Line189-191). In the present study, cost data were collected via an online questionnaire completed by 1,293 RA patients in CREDIT cohort enrolled from 152 hospitals across 29 Chinese provinces. These points are fully described in the revised manuscript (Materials and Methods, Paragraph3, Line 200-223).

- How were the patients selected (e.g., consecutively, randomly, or selectively)?

Response: Thank you for your suggestion. In revised manuscript, we added the subheading of “Target population and sample patients”, in which we described clearly. In short, the target population were consecutively recruited and the sample of cross-sectional study was the sample of convenient sampling according to the willingness to participate. (Materials and Methods, Paragraph3- Paragraph4).

- You need to state in the Methods section that you have followed STROBE guidelines: ‘The reporting of this study conforms to STROBE. (Insert new reference number)

Response: Thank you for your good suggestion. In the revised manuscript, the “Study design” of “Materials and Methods” subsection now states: “The study was reported in accordance with the STROBE guidelines.”and also marked the reference. (Materials and Methods, Paragraph1, Line187-189)

Results

- Abbreviations should be explained as subtitles below the Figures / Tables

Response: Thank you for the suggestion. Abbreviations were defined in footnotes beneath Tables 1, 3, and 4 of the revised manuscript.

Discussion

- A comparison of your results and the relevant previous studies should be made.

Response: Thank you for your good suggestion. The comparison of our results to previous studied in the mainland of China were presented in the discussion of revised manuscript. Furthermore, we explained the possible reasons for the differences. All these changes have been marked in the revised manuscript. (Discussion, Paragraph2, Line 450-454; Line 482-487)

- How can future research build on these observations? What are the key experiments that must be done?

Response: Thank you for your good suggestion. In the discussion, we tried to add a paragraph on the outlook for further search. That was “Based on the cost per patient estimated in our study, given the vast of RA patients and the trend of increasement of RA prevalence in China, the policy decision-makers and researchers need to recognize the increasingly severe economic burden associated with RA, which is progressively escalating and presenting a significant challenge to our society. To tackle these challenges, future research need focus on the following key areas: First, the sustained implementation of treat-to-target strategies for RA patients and the promotion of regional equity in treatment access are vital, as they might effectively reduce the economic burden of RA. Second, accelerating pharmacoeconomic studies is imperative. By leveraging market competition to lower drug prices, especially for biologics and targeted therapies, and through strategic negotiations with pharmaceutical companies, it is possible to expand the list of reimbursable RA medications and significantly reduce out-of-pocket costs for patients. These measures, combined with targeted adjustments to health insurance policies, can effectively mitigate the economic burden on individuals and families.” (Discussion, Paragraph4, Line 525-538)

Reviewers' comments:

Reviewer #1:

This manuscript addresses an important topic, providing valuable insights into the economic burden of rheumatoid arthritis (RA). The study design and methodological rigor are commendable, and the detailed cost analysis is a notable strength. However, some areas require further refinement to enhance clarity and impact.

Study Design and Methods: The recruitment flowchart is helpful, but the criteria for excluding "unqualified questionnaire quality" (n=22) need clarification. Additionally, the rationale for selecting the Gradient Boosting Machine (GBM) method over other approaches should be elaborated.

Response: Thank you for your comments. In response to the comment, the “Target population and sample patients” subsection now specified: “Following pilot testing and expert review, questionnaires completed in <3 min or with >50 % missing data were excluded. Finally, 12 patients were removed from our analysis.” in the revised manuscript. (Materials and Methods, Paragraph4, Line224-226).

We also have added a concise justification for using GBM: “As a tree-based integrated method, GBM could capture non-linearities and high-order interactions automatically, mitigate model misspecification through iterative gradient optimization, and yield smoothly calibrated probabilities that avert extreme propensity scores ( 0 or 1). This stabilized the inverse-probability weights and minimizes their undue influence on estimates”. (Materials and Methods, the subsection of “Statistical Analysis”, Line310-315).

Data Presentation: The cost distribution chart is effective, but the discussion on medication expenses could explore potential cost-containment strategies, such as the role of generics or insurance reforms. Consider condensing the frequency tables into supplementary materials for improved readability.

Response: Thank you for your good suggestion, we added the S2 Tables 2 including the more information in Supplementary material.

Ethical and Research Transparency: Describe in detail, if applicable, IRB approval, funding, and detailed information about data availability. Include a data availability statement to foster reproducibility.

Response: Thank you for your comments, we add the information about Data availability statement (Page33. Line 590-592), Ethics statement (Page33. Line 594-598), and Funding in the end of revised manuscript (Page33. Line 600-607).

Interpretation and Recommendations: Even though the study has identified a significant cost component, the discussion should be further extended to cover policy implications in particular, which may include indirect cost reduction strategies such as telemedicine or workplace accommodations. Future research ideas could include extending this framework to other conditions.

Response:

Thank you for your good suggestion. In response to the comment, we tried to add a paragraph in the discussion of the revised manuscript. That was “Based on the cost per patient estimated in our study, given the vast of RA patients and the trend of increasement of RA prevalence in China, the policy decision-makers and researchers need to recognize the increasingly severe economic burden associated with RA, which is progressively escalating and presenting a significant challenge to our society. To tackle these challenges, future research need focus on the following key areas: First, the sustained implementation of treat-to-target strategies for RA patients and the promotion of regional equity in treatment access are vital, as they can effectively reduce the economic burden of RA. Second, accelerating pharmacoeconomic studies is imperative. By leveraging market competition to lower drug prices, especially for biologics and targeted therapies, and through strategic negotiations with pharmaceutical companies, it is possible to expand the list of reimbursable RA medications and significantly reduce out-of-pocket costs for patients. These measures, combined with targeted adjustments to health insurance policies, can effectively mitigate the economic burden on individuals and families.” (Discussion, Paragraph4, Line 525-538)

In summary, this manuscript is a strong contribution to understanding the economic burden of RA. Addressing the above points will improve the study's clarity, robustness, and practical implications.

Reviewer #2: Reviewer Comments (Minor Revisions):

The manuscript presents a well-designed cross-sectional cost-of-illness study using data from a large national RA registry. The application of inverse probability weighting (IPW) via generalized boosted modeling and the use of bootstrap methods for cost estimation are appropriate and robust. The findings offer valuable insight into the economic burden of RA in China and identify key cost drivers that have clinical and policy relevance.

However, I would suggest two minor revisions to improve the clarity and interpretability of the study:

1. Survey Response Rate and Generalizability: The response rate of approximately 7% (1,293 of 18,507 eligible patients) raises potential concerns about non-response bias. Although IPW adjustment mitigates this to some extent, residual bias from unmeasured confounders (e.g., income, health-seeking behavior) may remain. I recommend briefly expanding the discussion on this limitation and its implications for the generalizability of the findings.

Response: Thank you for your good suggestion. We also agree with you that the low response rate might cause the potential non-response bias, so we added the relevant limitations in the discussion. That was “Lastly, considered the overall response rate of only 7 %, our cross-sectional study was susceptible to non-response bias. Although IPW mitigated imbalances in observed characteristics, residual confounding from unmeasured factors—such as income and health-seeking behaviors—might persist, potentially limiting the external validity of our findings to the wider RA population.” (Discussion, Paragraph 5, Line 562-567)

2. Attribution of Costs to RA vs. Comorbidities: It is not entirely clear whether the estimated costs are specific to RA-related healthcare utilization or may include expenditures related to comorbidities. Since RA patients often have overlapping medical conditions, further clarification in the Methods and Discussion sections regarding the attribution of costs would enhance transparency.

Response: Thank you for your comments. The costs estimated in our study included all the expenditures irrespective of their association with RA. So, it might overestimate. We further clarified in the Methods (Materials and Methods, Line240-242) and Discussion sections of revised manuscript (Discussion, Paragraph 5, Line 542-552).

These revisions are relatively minor and do not detract from the overall quality of the work. I support publication pending minor revisions.

Response: We sincerely appreciate your support.

Reviewer #3:

The research entitled "Average Annual Costs of Rheumatoid Arthritis Estimated by Inverse Probability Weighting and Their Predictors: A Cross-Sectional Study Based on the Chinese Registry of Rheumatoid Arthritis (CREDIT) Cohort" explicitly acknowledges a number of caveats and limitations.

1. Cross-Sectional Design: The study employs a cross-sectional approach, indicating that data were gathered at a singular moment in time. This methodological framework constrains the capacity to ascertain causal relationships between predictors, such as disease activity or treatment modalities, and annual expenditures.

Response: We appreciate the reviewer’s highlighting this important limitation. We fully agree that the cross-sectional design couldn’t achieve causal inference between predictors and annual costs; it can only describe associations at one point in time. In the revised manuscript we have therefore: (1) Revised the title: Average annual costs of Rheumatoid Arthritis estimated by inverse probability weighting and their influence factors: a cross-sectional study based on Chinese Registry of Rheumatoid arthritis (CREDIT) Cohort. (2) Clarified the study aim: Finally, we estimated annual per capita costs and their influence factors using the Bootstrap method. (Introduction, paragraph 5, Line 172). (3) Explicitly stated the limitation: “Fourthly, given the cross-sectional design, we couldn’t exclude the possibility of reverse causation or residual confounding. Therefore, the observed associations between the influence factors and expenditures of RA warrant confirmation in further prospective cohort studies.” (Discussion, paragraph 5, Line 559-562).

2. Data Acquisition and Representational Validity: Data pertaining to outpatient and inpatient expenditures were gathered through online questionnaires, a method that may potentially lead to recall bias or inaccuracies in reporting if patients fail to accurately remember or disclose their expenses.

Response: Thank you for this important observation. We agree with you that the estimation of costs was based on data collected by patient-self reported and there might be recall bias. In order to improve the accuracy as much as possible, we only invited the patients whose clinical records were updated by the clinicians who they visited during last 30 days, and participants were requested to recall the expense information of this recent visit in detail and frequency of visits last year. Moreover, at the initial stage, 53 patients were randomly selected to collect information repeatedly in two weeks to assess agreement of information. All of these were added in the discussion of revised manuscript. (Discussion, Paragraph 5, Line 554-559)

3. The sample was derived from the CREDIT cohort, and while inverse probability weighting (IPW) was employed to construct a weighted population that mirrors the larger RA patient demographic, there exists a possibility that the sample may not comprehensively represent

---

## [Decision Letter · Decision Letter 1]

30 Jul 2025

Average annual costs of Rheumatoid Arthritis estimated by inverse probability weighting and their influence factors: a cross-sectional study based on Chinese Registry of Rheumatoid arthritis (CREDIT) Cohort.

PONE-D-24-53064R1

Dear Dr. Wang,

We’re pleased to inform you that your manuscript has been judged scientifically suitable for publication and will be formally accepted for publication once it meets all outstanding technical requirements.

Kind regards,

Wesam Gouda, MD,PhD

Academic Editor

PLOS ONE

Reviewers' comments:

Reviewer's Responses to Questions

**Comments to the Author**

Reviewer #2: All comments have been addressed

Reviewer #3: All comments have been addressed

2. Is the manuscript technically sound, and do the data support the conclusions?

Reviewer #2: Yes

Reviewer #3: Yes

3. Has the statistical analysis been performed appropriately and rigorously?

Reviewer #2: Yes

Reviewer #3: Yes

4. Have the authors made all data underlying the findings in their manuscript fully available?

Reviewer #2: Yes

Reviewer #3: Yes

5. Is the manuscript presented in an intelligible fashion and written in standard English?

Reviewer #2: Yes

Reviewer #3: Yes

Reviewer #2: (No Response)

Reviewer #3: (No Response)

**Do you want your identity to be public for this peer review?** For information about this choice, including consent withdrawal, please see our Privacy Policy

Reviewer #2: **Yes: ** Yiduo Sun

Reviewer #3: **Yes: ** Adel Azzam

---

## [Editor Report · Acceptance letter]

PONE-D-24-53064R1

PLOS ONE

Dear Dr. Wang,

I'm pleased to inform you that your manuscript has been deemed suitable for publication in PLOS ONE. Congratulations! Your manuscript is now being handed over to our production team.

Kind regards,

on behalf of

Dr. Wesam Gouda

Academic Editor

PLOS ONE